# How We Found Our IMU: Guidelines to IMU Selection and a Comparison of Seven IMUs for Pervasive Healthcare Applications

**DOI:** 10.3390/s20154090

**Published:** 2020-07-22

**Authors:** Lin Zhou, Eric Fischer, Can Tunca, Clemens Markus Brahms, Cem Ersoy, Urs Granacher, Bert Arnrich

**Affiliations:** 1Digital Health Center, Hasso Plattner Institute, University of Potsdam, 14482 Potsdam, Germany; eric.fischer@student.hpi.uni-potsdam.de; 2NETLAB, Department of Computer Engineering, Bogazici University, 34342 Istanbul, Turkey; can.tunca@boun.edu.tr (C.T.); ersoy@boun.edu.tr (C.E.); 3Division of Training and Movement Sciences, University of Potsdam, 14469 Potsdam, Germany; mbrahms@uni-potsdam.de (C.M.B.); urs.granacher@uni-potsdam.de (U.G.)

**Keywords:** inertial measurement unit, pervasive healthcare, gait analysis, comparison of devices

## Abstract

Inertial measurement units (IMUs) are commonly used for localization or movement tracking in pervasive healthcare-related studies, and gait analysis is one of the most often studied topics using IMUs. The increasing variety of commercially available IMU devices offers convenience by combining the sensor modalities and simplifies the data collection procedures. However, selecting the most suitable IMU device for a certain use case is increasingly challenging. In this study, guidelines for IMU selection are proposed. In particular, seven IMUs were compared in terms of their specifications, data collection procedures, and raw data quality. Data collected from the IMUs were then analyzed by a gait analysis algorithm. The difference in accuracy of the calculated gait parameters between the IMUs could be used to retrace the issues in raw data, such as acceleration range or sensor calibration. Based on our algorithm, we were able to identify the best-suited IMUs for our needs. This study provides an overview of how to select the IMUs based on the area of study with concrete examples, and gives insights into the features of seven commercial IMUs using real data.

## 1. Introduction

Inertial measurement units (IMUs) are electronic devices that typically consist of a 3-axis accelerometer (which measures linear acceleration) and a 3-axis gyroscope (which measures angular velocity). Some IMUs also contain a 3-axis magnetometer (measuring magnetic field), which are specified as MIMUs. By fusing information from these sensors, the orientation of the IMU can be estimated [1]. IMUs have been gaining popularity in the past years as a research tool for pervasive healthcare [2,3]. On the other hand, the selection of an IMU suitable for the research topic is also increasingly challenging due to the large variety of options. The following paragraphs will provide a brief introduction of the commercially available IMUs, their applications in pervasive healthcare related to gait, and illustrate how the current study can contribute to the knowledge of IMU selection for pervasive healthcare related use cases.

### 1.1. Commercial IMUs for Pervasive Healthcare

IMUs are commonly used in applications where navigation or body movement tracking is needed, such as aviation, industrial robotics, as well as indoor navigation or human motion tracking. The angular velocity and acceleration measured by the IMU is often integrated to obtain information about orientation or displacement. During the integration process, small errors and other noises from the raw data can lead to large drifts in the calculated results. Thus, IMU data analysis is susceptible to even very small errors. Titterton and Weston described in detail models to measure the errors of gyroscope and accelerometer [4]. The contribution of the errors from the sensors in an IMU after calibration can be used to describe its performance. Ranging from high- to low-performance, commercially available IMUs can be categorized into the following performance classes; (1) marine and navigation grade IMUs, which are typically used on ships, spacecrafts, or aircrafts; (2) tactical-grade IMUs, which are commonly used for unmanned aerial navigation, (3) industrial-grade IMUs, which are used for industrial robotics and machinery; and (4) hobbyist, automotive, or consumer grade IMUs, which are commonly used for wearables, virtual reality or gaming. Despite the lack of strict definitions to classify IMUs into the above-mentioned performance classes, Chao et al. provided an example range of price, weight, gyroscope-, and accelerometer bias for IMUs in all four performance classes [5]. There is a trade-off between the performance of the IMUs and other characteristics, such as their size or price. Therefore, when searching for IMUs, researchers should first identify the performance requirements for their research areas.

Pervasive healthcare is among the most prominent fields of research with an increasing demand for IMUs. Typically, IMUs are used to track movement patterns or recognize activities of the user. The information extracted from the IMU data can then be used for advising treatments or motivating behavioral changes. For example, IMUs can be used to measure medication dosage response and facilitate individualized therapy for Parkinson’s patients by monitoring muscle tremor [6], to track unwanted stereotypical movements in daily and provide real-time feedback [7], or to estimate the amount of physical activity from the user, thus inferring the energy expenditure and provide reference for lifestyle changes [8,9]. Research topics in this field are often oriented towards daily life applications, and the IMU data is mostly processed to gain insights into the human activities. In a great majority of cases, the performance requirements of the IMUs are lower compared to those of industrial or tactical grade IMUs. For the purpose of this paper, we limit the scope of discussion to IMUs that can be roughly categorized into consumer grade.

The growing market and technological advancements in recent years have led to the emergence of large varieties of integrated IMUs. In these IMUs, the sensors, battery, and communication modules are readily integrated. Additionally, many of these products also have accompanying software or apps, making them the out-of-the-box solutions for routine data collection. A large number of studies have evaluated the performance of individual IMU models for their applications in human motion tracking and healthcare [10,11,12]. However, the large diversity of commercially available IMUs and the different evaluation procedures in existing studies makes it increasingly difficult for researchers to select an IMU that is best-suited for their needs. To make matters worse, as the exact data collection pipeline as well as the data quality only become clear with hands-on experience, it is hard to identify and compare relevant features of different IMUs beforehand. In typical studies using IMU data, it is not common to see descriptions about how or why the particular IMUs were selected. Researchers who have the freedom to choose their own IMUs for a study will likely spend considerable time researching the available IMUs on the market. Moreover, apart from the advertised features, very little information is available about how well a particular IMU will perform in the context of a specific use case.

Therefore, the current study provides practical guidelines for IMU selection in the field of pervasive healthcare (i.e., gait analysis). To give a concrete example of the selection procedure, and to show the considerations of IMU features according to a specific use case, we present the results of a comparative study including seven frequently used IMUs by applying the workflow laid out in the guidelines. For researchers who are interested in using one or a few of the seven IMUs for similar use cases, the results are directly informative. Based on the structured selection procedures and considerations suggested in the guidelines, researchers can also define a different set of IMU candidates, and customize the selection criteria based on requirements for their specific research topics.

### 1.2. Gait Analysis with IMUs

Gait analysis is the study of walking patterns. Being the fundamental of human locomotion, gait is an important indicator of a person’s cognitive and physical health [13,14]. Clinical gait analyses are routinely conducted by medical professionals in the diagnosis, treatment, or rehabilitation process of patients with neurological diseases, such as Parkinson’s disease or stroke [15,16].

Human gait can be quantified using a set of spatio-temporal gait parameters, such as stride length, stride time (gait cycle time), clearance (maximum elevation of the foot), stance/swing time, speed, and cadence. Figure 1 illustrates the human gait cycle, its major events, and its phases. Gait speed has been termed the sixth vital sign and has been reported to correlate with mortality [14,17]. Stride-to-stride variability of stride time, stride length, as well as swing time are examples of valid indicators for falls risk [18,19,20]. Recent studies showed that the gait characteristics in controlled lab settings and daily life settings are different [21,22]. Therefore, being able to monitor gait patterns in an unobtrusive and cost-effective manner in real-life situations is of great interest to the goal of pervasive healthcare. Consumer-grade IMUs are commonly used for such purposes. They have proved to generate valuable insights [23,24], and are applicable for large cohorts [25].

Estimating spatio-temporal gait parameters from IMU data is not a trivial task because of the IMU drift problem. Common strategies for accurate calculation of the gait parameters include (1) model-based approaches, where the human leg is modeled as a double pendulum. By combining information of the stride length with the angular velocity of leg swing, spatio-temporal gait parameters can be derived [26,27]. (2) Machine learning approaches, where the spatio-temporal gait parameters are directly predicted using a set of features extracted from the IMU signal [28,29]. (3) Double-integration-based sensor fusion approaches, where the laws of physics are applied by integrating acceleration two times into displacement [30,31,32]. Despite attempts in recent years to make the machine learning approaches more explainable [33], the double-integration-based approach enables realistic and direct reconstruction of 3D foot trajectories, which makes the gait analysis results more understandable and interpretable compared to the other two mentioned above. However, the data analysis procedures for this approach are sophisticated. For example, in the algorithm developed by Tunca et al. [34], accumulated errors in IMU data were tracked and corrected by an error state Kalman filter using zero-velocity update; foot orientation was estimated by a particle filter, which helps to distinguish turning steps from those during straight walk; and initial contact and foot off time points were then determined based on the signal features. The accuracy of gait parameter estimates produced by this algorithm was validated [34]. Nevertheless, each step of the calculation relies heavily on the raw data quality, and small errors from different IMUs could be magnified in the final results, leading to less accurate gait parameter estimates. Thus, in the current study, this gait analysis algorithm was used to analyze the data from all tested IMUs, and the accuracy of the estimated spatio-temporal parameters were compared to a gold standard in order to gain insights into the raw data quality of different IMUs.

### 1.3. Structure of the Paper

In this paper, we first propose guidelines for selecting IMUs for pervasive healthcare related studies. Afterwards, we describe the comparative analysis of seven different IMUs following the steps described in the guidelines. First, the IMU device specifications were collected and listed, then the data collection procedures and raw data were explored. Finally, using gait analysis as an example use case, we compare the accuracy of the spatio-temporal gait parameters calculated with raw data from the seven IMUs. In the discussion section, typical use cases of IMUs in pervasive healthcare, as well as the corresponding requirements, are highlighted.

## 2. Proposed Guidelines for IMU Device Selection

In this section, we propose a guideline for selecting IMU devices. The general procedures including three major steps are illustrated in Figure 2. The “Initial Selection” step is collecting and confirming specifications about the IMUs, all of the information described in this step can be obtained before acquiring the actual IMU devices. The specifications of the sensors as well as the relevant information about the products can play a crucial role in filtering for the IMUs. Table 1 shows the recommended list of specifications and their relevance in IMU selection for pervasive healthcare use cases.

Most of the IMUs have acceleration and gyroscope range of ±16 g and ±2000 deg/s, respectively, and sampling rates up to a few hundred Hz. A sampling rate around 100 Hz is sufficient for capturing daily life human activities, such as walking, or picking up objects. For example, Payton et al. recommended 25–50 Hz for general gait analysis [35]. Apart from the type of activities, the requirement of sampling rate also depends on the type of analysis. For example, classification of activities usually requires lower sampling rates compared to high precision movement trajectory reconstruction [36,37]. Using a sampling rate higher than necessary will increase the computation load without improving the results. In a study on classification of walking conditions, the same algorithm achieved similar performance on data with sampling rates ranging from 25 Hz to 400 Hz [36]. Onboard memory and battery capacity should be considered for real-life, long-term data collection. In pervasive healthcare use cases, the subjects are likely to be recorded while moving freely in daily life environments. Due to the large movement domain and signal occlusions in buildings, the wireless data transmission might be unstable at times. The data loss would lead to inaccurate results in further analysis. Battery capacity could be a limitation for long-term recordings. However, for devices that offer developer options, it is also possible to develop automated data transfer protocols that save the data in a separate storage place on a regular basis, thus enabling extended data collection period. Charging option could be highly relevant for home use cases. For example, in a long term study where Parkinson’s disease patients are asked to charge their own devices during the data collection period, wireless charging is much more manageable than Micro USB charging. Additional sensors such as ambient light sensor, barometer (air pressure sensor), and thermometer (temperature sensor), could be useful for user environment recognition. For example, determining if the user is indoor or outdoor, or if the user is moving upstairs or downstairs in a building. Water/dust proof can be a useful feature to expand the choice of daily life scenarios for data collection.

Many of the IMU devices also provide onboard sensor fusion, which uses raw acceleration and angular velocity data to calculate orientation, either as quaternions or Euler angles, in almost real-time. Depending on the use case, this feature is not always necessary. For algorithm development, the researcher is advised to work directly with the raw data and implement sensor fusion as part of the algorithm, in order to have more control over the data processing procedures. When multiple devices are used for data collection, clock synchronization, and multi-sensor control can significantly reduce the amount of time used for data recording and for data processing. In cases where the recorded data is not synchronized, a practical strategy is to shake the devices simultaneously to produce distinct movement signatures in the raw data, which can then be used to align the entire data recording period [38]. There are three major types of solutions of calibration offered by the IMU devices: (1) a calibration status category (e.g., “good”, “acceptable”, and “not calibrated”) is being streamed to the app or software, and the user rotates the IMU in different orientations until the calibration status indicates high accuracy. (2) The IMU is calibrated with a dedicated calibration software, which requires more rigorous calibration procedures and calculates the calibration parameters to be written to the devices. (3) The IMU was pre-calibrated in the factory and do not require further calibrations by the user. The special features from each device can play an important role for the selection. For example, live visualization of sensor orientation offers the opportunity to check in (almost) real-time the sensor fusion behavior and is a practical tool for demonstrations. Some IMU devices also have companion software or apps designed for data analysis dedicated use cases, such as activity triggered questionnaires or clinical motor function tests.

With the above-mentioned considerations, a shortlist of IMUs that are potentially suitable for the intended study can be identified. However, to gain insights into the use case specific information, it is strongly recommended that example devices are obtained for the following two steps of the guideline. Apart from purchasing new devices, researchers may have the opportunity to borrow or rent a few devices for testing purposes. The “Explore Raw Data” step aims at gaining hands-on experience with the data collection workflow, as well as exploring the quality of the raw data, thus detecting potential issues for the intended use case at an early stage. Table 2 summarizes recommended aspects to be checked during initial data recordings.

The recording duration and number of devices are major factors that influence the preference for data recording control. Simultaneous control for multiple devices on a smartphone is especially convenient for short term recordings (e.g., a few minutes), as the data can be collected and then analyzed quickly. Long-term recordings (e.g., days) with one device can be configured via cable on a PC; this might take more time, but is usually more robust against data loss. In addition, the download speed is much faster than Bluetooth download for large size data. Some devices might require that the previous recording session is downloaded before the next recording session can start. This could be a measure to secure the recorded data, but on the other hand, it can be also time consuming. Apart from getting experience with the data collection workflow, some tests with the data can also help gaining insights that are relevant for the intended use case. Recommended first tests include checking the maximum recording time, exported timestamps, and the baselines of the raw data. Details of these procedures are described in the Section 3.2.

After collection of device specifications and exploration of raw data, the next step is to test the IMUs with a use case algorithm. It is worth emphasizing that the results of the use case algorithm testing depend both on the features of the IMU device itself, and on the data processing procedures in the algorithm. IMUs from different producers have different error characteristics, which may or may not affect the results of certain types of analysis. Therefore, it is highly recommended that the algorithm is the same as the one used in the intended study with the IMUs, or highly resembles the processing procedures for the IMU data in the intended study. Depending on study specific requirements, the researcher should pay equal attention to the results of all three major steps in the guideline. Nevertheless, the result in the use case algorithm testing can serve as a reliable measure of the most suitable IMUs, as it combines all factors from the IMU and the data processing, and gives a direct indication of how the IMUs will perform in the real study.

## 3. Materials and Methods

### 3.1. Devices

#### 3.1.1. Device Inclusion/Exclusion Criteria

Commercial IMU devices were screened for their potentials for pervasive healthcare-related applications. In particular, they had to meet all of the criteria described in Table 3 in order to be included in the current study. We chose to only include off-the-shelf commercial IMUs with integrated battery, memory, and Bluetooth modules in our analysis, because this study targets at researchers who prefer to use existing data collection pipelines of the IMUs. Independent IMUs which are only composed of an accelerometer and a gyroscope were excluded from the study. Examples of excluded IMUs for this reason are MPU6050 (InvenSense, San José, CA, USA), BMI160 (Bosch, Gerlingen, Germany), and BMI055 (Bosch, Gerlingen, Germany). It is worth noting that these independent IMUs can also be part of an integrated commercial product. For example, the BMI160 IMU is also the IMU component in the MetaMotion devices (MbientLab, San Francisco, CA, USA). Researchers who are selecting IMUs from both categories should be aware of the possible redundancy. Accelerometer and gyroscope were considered a minimum requirement of the IMUs, as the combination of these data sources will give a more complete description of the movement in terms of rotation and translation, thus enabling analysis on the movement trajectories. Motion tracking devices with only accelerometer or gyroscope are excluded from this study. For example, the activity monitor ActiGraph wGT3X-BT (ActiGraph, Pensacola, FL, USA), which only contains accelerometer, was excluded from the study. It is required for all devices tested in this study to have onboard memory storage, which enables the subjects to move freely without the restrictions from the range of wired- or wireless communication for data transmission, and without the problem of signal occlusion in buildings, particularly when assessing gait in facilities such as a senior residence. Data loss due to unstable wireless transmission would lead to inaccurate results in further analysis. With onboard recording, the data can then be downloaded from the devices after the recording sessions. IMUs from the company Xsens (Xsens, Enschede, The Netherlands) are commonly used for human motion capture, and the reliability and concurrent validity of the measurements have been proven [10]. However, the Xsens IMUs were excluded from the current study because to our knowledge, they are not able to record data onboard. Depending on the area of research, raw data access can be an essential requirement for gaining more insights into the human activity. Smartwatches and fitness trackers have been gaining popularity in recent years as research tools to track physical activity [39]. However, researchers should be aware that many of the device models only provide processed measurements, such as step count or sleep quality, but do not permit access to raw IMU data. Example devices with restrictions on raw data access include Steel HR (Withings, Issy-les-Moulineaux, France) and MiBand (Xiaomi, Hangzhou, China). The current study focused on comparing devices with dedicated IMU functionalities and did not explicitly consider smartwatches and fitness trackers. The cost of the devices was also taken into consideration, as some studies might require large quantities of them for data collection in parallel. Based on preliminary market research, the price threshold for inclusion criteria in the current study was set at 600 €. IMU models that are designed for full body motion capturing in studios or specialized for analyzing sport performance typically cost much more than average single IMU devices. Examples of IMUs in the high-cost category that were excluded in the current study are APDM OPAL (APDM, Portland, OR, USA), Blue Trident IMU (Vicon, Oxford, UK), and Perception Neuron Studio Inertial System (Noitom, Miami, FL, USA).

#### 3.1.2. Devices and Configurations Used in the Current Study

Based on the selection criteria, as well as their availability, the following IMU devices were selected for comparison in this study; QuantiMotion (Bonsai Systems, Zurich, Switzerland), MetaMotionR (MbientLab, San Francisco, CA, USA), NilsPod (Portabiles, Erlangen, Germany), Move 4 (movisens, Karlsruhe, Germany), Physilog®5 (Gait Up, Lausanne, Switzerland), EXL-s3 (EXEL, Bologna, Italy), and Shimmer 3 (Shimmer Research, Dublin, Ireland). If the device is the main IMU product of the company, the name of the company rather than the name of the device is used in the following text. Table 4 summarizes the devices selected for testing in this study and their sampling rates used for data collection. The sampling rates were selected based on the available options that are closest to 100 Hz for each IMU. This sampling rate is sufficient for most types of human motion tracking, and is commonly used for human gait analysis studies which are similar to the example use case [32,37]. The IMUs were configured to record with accelerometer and gyroscope using their mobile apps whenever possible. More specifically, the Bonsai and MMR IMUs were configured using their iOS apps, the Portabiles, GaitUp, and Shimmer IMUs were configured using their Android apps, and the Move 4 and EXLs3 IMUs were configured using their PC applications. The accelerometer range was ±8 g (except for Bonsai IMUs, whose acceleration limit was ±4 g), and gyroscope range was ±2000 deg/s. The raw data files were downloaded either via Bluetooth, or transferred to the PC via a micro USB cable, and were all converted into a unified .csv format. Descriptions on options for sensor configuration and data collection control for all tested IMU devices can be found in Section 4.1.

### 3.2. Explore Data Collection Procedures and Raw Data

The following tests and analysis were performed to provide first insights into the data collection procedures and the quality of the raw data.

#### 3.2.1. Calibration and Preprocessing

Calibration was performed only for IMUs that have calibration options as part of the product. More specifically, the Bonsai and MMR IMUs were calibrated using the calibration status displayed in their smartphone apps, and the Shimmer IMUs were calibrated using the Shimmer 9-DoF calibration software. In this study, the IMUs were presented as newly purchased products. By minimizing the influence of custom calibrations, the data analysis results should be more informative in terms of the initial calibration status of the IMUs. Therefore, no additional custom calibration was performed for the IMUs. Researchers who are interested in more sophisticated calibrations can refer to the procedures described by Ferraris et al. [40]. Timestamps (if applicable), acceleration values, and gyroscope values from *x*-, *y*-, and *z*-axes, respectively, were extracted from the raw data files, and converted into the same format for further analysis.

#### 3.2.2. Maximum Recording Time

Two IMU devices from each selected brand were configured to record accelerometer and gyroscope signals onboard (except for EXLs3, which was in streaming mode) at sampling rates of 100 Hz, 102.4 Hz, or 128 Hz, depending on the sensor specifications. The devices were left to record data at a stationary position until the recordings automatically stopped, either due to full memory or empty battery. The data was then retrieved, and the maximum recording duration was documented.

#### 3.2.3. Timestamps

Timestamps from the recorded data were then examined for the following aspects; (1) whether the timestamps exist in the raw data output, (2) whether the timestamps were in Unix time format or time after the start of the recording, and (3) how accurate was the experimental sampling rate compared to the configured sampling rate. For IMUs that provide timestamps in the raw data, empirical sampling rates were calculated as the inverse of the time duration between adjacent timestamps:(1)$$f_{s}=\frac{1}{t_{i}-t_{i-1}}$$
where $f_{s}$ is the empirical sampling rate for a particular time interval, $t_{i}$ is the *i*th timestamp, and $t_{i-1}$ is the previous timestamp recorded in the raw data. The empirical sampling rates were then plotted as histograms to show the distributions around the expected sampling rate.

#### 3.2.4. Baseline Accelerometer and Gyroscope Values

A simple analysis on the raw data was then performed to check the calibration status of the sensors: acceleration and angular velocity values were recorded when the device was in stationary status. Two IMU devices from each tested brand were used for data collection. Five-thousand sample points were taken from each tested device, and the data distribution was illustrated with box plots. For perfectly calibrated sensors, the acceleration magnitudes should be exactly 1 g (gravity), and the angular velocity should be zero. However, these values are necessary but not sufficient for characterizing the sensor errors. Researchers interested in assessing raw IMU data on low-level (e.g., quantization noise, random walk noise, and bias instability) can refer to the Allan variance as a suitable method. It models random-drift errors over time by quantifying the frequency stability subsequently for two-sample variances. This can be used, for example, to compute the bias stability over time of a gyroscope [41].

### 3.3. Gait Analysis Experimental Set-up

To further evaluate the quality of the data recorded by the tested IMUs, IMU-derived spatio-temporal gait parameters were compared to a gold standard for gait analysis.

The OptoGait system (Microgait, Italy) was used as a gold standard to cross-validate IMU devices. The system consists of two 10 m bars which are placed on ground level in parallel. One of the bars transmits infrared signals, and the other one receives the signals. Once the recording has been initiated, the system detects signal interruption with high temporal- (1000 Hz) and spatial (1.04 cm) resolution caused by the subject’s foot movement while walking between the bars. It then outputs spatio-temporal gait parameters, such as stride length, stance and swing times, and walking speed. The spatio-temporal gait parameters measured by the system have been validated in several studies. For example, the gait parameters assessed with OptoGait correlated highly with a pressure sensitive gait analysis system (GAITRite^®^, CIR Systems, Inc., Franklin, NJ, USA), with intraclass correlation coefficients ranged between 0.933 (swing time) and 0.999 (cycle time, cadence, and walking speed) [42]. High concurrent validity has been determined between the OptoGait and the Vicon motion capturing system, with ICCs ranging from 0.690 and 0.999 (*p* < 0.001) [43].

Five young, healthy subjects were recruited for data recording for each of the seven tested types of IMUs, the characteristics of the subjects are described in Table 5. All of the subjects completed the following three testing sessions: Session 1—small strides, where the subject had to make sure that the feet did not overlap in the anteroposterior direction during double-support phase, as this could cause problems for the OptoGait foot detection mechanism. Session 2—normal strides. Session 3—large strides, where the subject was asked to maintain double-support phase (not running). Apart from the restrictions described above, the subjects were free to walk at their preferred speed and chose their natural gait patterns. Recordings from one subject for EXLs3 were not used for analysis due to data loss. The effects of data loss on gait analysis results are presented in Section 4.2. In each recording session, the IMUs were fixed on the top of the left and right shoes of the subject. Upon the instruction of the researcher, the subject walked back and forth between the OptoGait bars six times (resulting in a total distance of 60 m) and stopped outside of the OptoGait area. Whenever applicable, the study design was compliant with the guidelines for clinical applications of spatio-temporal gait analysis proposed by the European GAITRite^®^ network group [44]. The study was approved by the ethics committee of the University of Potsdam (29/2020), and all experiments were conducted according to the latest revision of the declaration of Helsinki. All subjects gave their consent before participating in the data collection procedures. Figure 3 shows the experimental set-up.

### 3.4. Data Analysis

This section describes the data analysis procedures for deriving the spatio-temporal gait parameters from the IMU data, and assessment of the spatio-temporal gait parameters. The spatio-temporal gait parameters were calculated using Matlab R2019b (The MathWorks Inc., Natick, MA, USA), the evaluation of the gait parameters and all other data processing procedures in this study were performed using Python 3.7.4 (Python Software Foundation, Wilmington, DE, USA).

#### 3.4.1. IMU Gait Analysis Algorithm

Spatio-temporal gait parameters were calculated from the raw IMU data using a gait analysis algorithm developed by Tunca et al. [34]. The algorithm utilizes the well-known zero-velocity update technique to cope with the drift errors due to the double integration of acceleration measurements [30,45]. In addition, it uses an error-state Kalman filter to further correct the displacement estimates. Specifically, the acceleration and angular velocity measured by the IMU are represented by
(2)$$f_{t}=\left(f_{t}^{x},f_{t}^{y},f_{t}^{z}\right)$$
(3)$$\omega_{t}=\left(\omega_{t}^{x},\omega_{t}^{y},\omega_{t}^{z}\right)$$
where $f_{t}$ and $\omega_{t}$ are the accelerometer and gyroscope output measured at time *t*, respectively. The state of the IMU is defined by its attitude, velocity, and displacement:(4)$$x_{t}=\left(R_{t},v_{t},s_{t}\right)$$
where $R_{t}$ is the rotation matrix representing the attitude of the IMU in the global coordinate frame, and $v_{t}$ and $s_{t}$ are 3-dimensional vectors specifying its velocity and displacement, respectively. The attitude $R_{t}$ is estimated with the following integration scheme,
(5)$$R_{t}=R_{t-1}\left(I+\frac{\sin\sigma }{\sigma }B_{t}+\frac{1-\cos\sigma }{\sigma^{2}}B_{t}^{2}\right)$$
where I is the identity matrix, $\sigma =|\omega_{t}\delta t|$, where δt is the time between consecutive measurements, and $B_{t}$ is the skew-symmetric matrix form of the vector $\omega_{t}$. The velocity and displacement estimated at time *t* starting from their values at time t−1 are then calculated by
(6)$$a_{t}=R_{t}f_{t}+g$$
(7)$$v_{t}=v_{t-1}+\delta t\cdot a_{t}$$
(8)$$s_{t}=s_{t-1}+\delta t\cdot v_{t}$$
where g is the vector denoting gravitational acceleration, and $a_{t}$ is the acceleration estimate with gravity excluded.

These equations are sufficient to track the state of the IMU under perfect error conditions, but in practice, the noise and bias in IMU measurements cause drift which have to be regularly corrected, for which a Kalman filter is employed. The filter has two alternating stages: prediction and correction. In the prediction stage, the error covariance $P_{t|t-1}$ is estimated by
(9)$$F_{t}=\left(\begin{array}{ccc} I & 0 & 0\\ \delta tS_{t} & I & 0\\ 0 & \delta tI & I \end{array}\right)$$
(10)$$P_{t|t-1}=F_{t}P_{t-1|t-1}F_{t}^{T}+\Sigma_{w}$$
where $S_{t}$ is the skew-symmetric matrix form of the vector $R_{t}f_{t}$, 0 is a zero matrix, $F_{t}$ is the error-state update matrix, and $\Sigma_{w}$ is the noise covariance matrix modeling the sensor errors. The correction stage is executed during the zero-velocity phase where the foot is determined to be completely stationary and is in full contact with the floor
(11)$$\begin{array}{cc} K_{t} & =P_{t|t-1}H^{T}{\left(HP_{t|t-1}H^{T}+\Sigma_{v}\right)}^{-1} \end{array}$$
(12)$$\begin{array}{cc} \delta {\hat{x}}_{t} & =K_{t}\left(0-H{\hat{x}}_{t|t-1}\right) \end{array}$$
(13)$$\begin{array}{cc} P_{t|t} & =P_{t|t-1}-K_{t}HP_{t|t-1} \end{array}$$
where $K_{t}$ is the Kalman gain, $H=\left(\begin{array}{ccc} 0 & I & 0 \end{array}\right)$ is the 3×9 matrix filtering the state so that only the velocity elements remain non-zero, ${\hat{x}}_{t|t-1}$ is the predicted state calculated via Equations (5)–(7), and $\Sigma_{v}$ is the error covariance matrix modeling the error in the zero-velocity updates. The corrected state is then calculated by
(14)$${\hat{x}}_{t}={\hat{x}}_{t|t-1}+\delta {\hat{x}}_{t}$$

As a result, 3-dimensional displacement of the feet ($s_{t}$) are computed, which can then be converted to spatial gait parameters such as stride length. Primary gait events (i.e., initial contact and foot off events) were detected using a peak detection-based method on the raw angular velocity measurements. Consequently, the gait phase durations, such as stance, swing and cycle times, could be calculated.

#### 3.4.2. Assessment of the Spatio-Temporal Gait Parameters

Turning steps made by the subject outside the OptoGait area were excluded from our analysis. The strides made during straight walking at both ends of the OptoGait system were also discarded, as these strides could lay partially outside of the OptoGait, causing a shorter stride measurement from the OptoGait system. To synchronize the IMU and the OptoGait recordings, the subjects were instructed to stand still for a few seconds outside of the OptoGait walkway, and always initiate walking by stepping into the OptoGait walkway with the right foot. In this way, the timestamps of initial contact of the very first stride could be identified in both IMU and OptoGait recordings and used for synchronization. The subsequent strides calculated by the IMU gait analysis algorithm and the strides detected by the OptoGait system were then matched based on timestamps of initial contact. The stride lengths as well as the stride times calculated from the gait analysis algorithm were then compared with those measured by the OptoGait system. Linear regression parameters such as correlation coefficient, slope and intercept of the regression line, and root mean square error were obtained from correlation analyses, and limits of agreement at ±1.96 standard deviation were obtained from Bland–Altman plots.

## 4. Results

### 4.1. Comparison of IMU Specifications

#### 4.1.1. Sensor Specifications

Table 6 summarizes technical specifications about the devices which are relevant for IMU selection. Almost all IMUs have an acceleration range of ±16 g and a gyroscope range of ±2000 deg/s, except that the default sampling rate that can be configured in Bonsai app is ±4 g, but customization to ±16 g is possible depending on the use case. The calibration status for MMR can only be displayed in a custom app, which the user will have to build according to the example scripts provided in the online tutorial. For the Shimmer devices, multi-device control as well as sensor fusion algorithm are only available with a paid version, based on an annual subscription fee.

#### 4.1.2. Data Collection Procedures

Table 7 summarizes data collection procedures for each device, including sensor configuration, how to stop recording, data download, and data format. The configuration, as well as recording control of most sensors, can be performed with a smartphone App. Move 4 devices only communicate with their Windows software on a PC, where it is possible to set up delayed recording for a predefined interval. With the onboard recording mode, no connection to a phone or PC is required, which allows for a large range for the movement. Bonsai and Gait Up devices offer live view of the device orientation in their apps, which requires the devices to stay within the range of the Bluetooth connection of the phone. For Bonsai devices, it is possible to record either onboard or stream data while keeping the live visualization; on the other hand, for Gait Up devices, no data will be recorded in the live visualization mode.

Recording for all devices that connect to a smartphone can be stopped in the app, the recorded data is also saved in the app and can be transferred using the phone’s file sharing options. For MMR, the current recording has to be downloaded via Bluetooth before the next recording session can start, which can be time consuming in case of long recordings. Similarly, the recording will stop when the Move 4 devices are connected to the PC, and the current recording has to be either downloaded or discarded before the next recording can start. Physical buttons on the Gait Up and Shimmer devices allow independent data recording control without communication with any additional hardware. However, the button on Shimmer could be tricky to handle as the haptic feedback is very weak.

#### 4.1.3. Raw Data Explorations

Five IMUs (Bonsai, MMR, Gait Up, EXLs3, and Shimmer) recorded timestamps for the samples in raw data. To investigate the accuracy of the sampling times, a random recording session with 5000 sample points was selected for each IMU. The real sampling rate calculated from each time interval between adjacent data points was calculated and plotted as a histogram, shown in Figure 4. The expected intervals are slightly different for the IMUs due to variations in the allowed sampling rates (100 Hz or 128 Hz). The timestamp intervals for Bonsai data are closest to the configured value, whereas the timestamp intervals for the EXLs3 data spread across a wide range surrounding the configured value (100 Hz). This was to be expected, as only the EXLs3 data was streamed via Bluetooth instead of recorded onboard. It is known that data packets can be lost during Bluetooth transmission, especially when streaming at high frequencies and with long data transmission distance. It was suspected that the timestamps were recorded at times data packet arrival, not times of data recording, as the large number of intervals close to zero indicated that these data packets probably arrived at the same time due to delays in data transmission.

The IMU devices were then tested for the baseline values of acceleration and angular velocity when the devices were placed at a stationary position (e.g., lying on the floor). Figure 5 and Figure 6 show the distribution of baseline values, two devices from each brand were used for data collection, 5000 sample points from each IMU were used for plotting. It could be observed that the baseline values vary among IMUs from different brands, and even between two IMUs from the same brand. It should be emphasized that the baseline analyses illustrated here are only initial explorations, whether the baseline values are acceptable or not depends heavily on the requirements of the use case.

### 4.2. Comparison of IMU Data Quality Using a Gait Analysis Algorithm

Correlation analysis and Bland-Altman plot were used to compare the spatio-temporal gait parameters calculated from the IMUs and those measured by the reference system. Table 8 and Table 9 summarize the information from correlation analysis and Bland-Altman plot for each IMU device for stride length and stride time. Correlation plots and Bland-Altman plots for stride length and stride time derived from all seven tested IMUs can be found in Appendix A. It is worth noting that EXLs3 IMUs were the only ones that streamed data via Bluetooth, which is prone to data loss, all other IMUs stored data onboard during recording. Data from one subject recorded with EXLs3 had to be discarded because of significant data loss (only around 80 % of data packets were retrieved), and the calculated strides could not be used for further analysis. Figure 7 shows a comparison of calculated stride trajectories from two EXLs3 recording sessions, trajectories from data with too many missing sample points were disrupted.

In terms of stride time, data from all IMUs produced similarly accurate results. The correlation coefficient and root mean square error (RMSE) indicated that stride times derived from the IMU data correlated closely with the measurements from the OptoGait system. Only for the EXLs3 IMUs, the slope of the linear regression line was smaller, and the intercept was larger for the compared to the other IMUs. Analysis with the Bland-Altman Plot also indicated high levels of agreement between the stride time estimations from all tested IMUs with the OptoGait system, with around 95% of the data points (±1.96 standard deviation) deviating less than 0.1 s from the OptoGait measurements. This was to be expected, as the temporal parameters were extracted directly from raw IMU data; with a sampling rate at or over 100 Hz, the gait events could be reliably identified. The correlation analyses and Bland-Altman plots for the above described IMU data can be found in the Appendix A (Figure A1 and Figure A2).

The accuracy of stride length estimations from different IMU data showed more variability. Data from Move 4, Gait Up and EXLs3 IMUs produced comparably accurate results, with high level of correlation (indicated by correlation coefficient of 0.99 and RMSE of 0.04 m), as well as high level of agreement (indicated by average limits of agreement less than 0.09 m) with the OptoGait system. Stride lengths estimations from other IMUs showed issues that could mostly be traced back to the raw data quality. For the Bonsai IMUs, the stride lengths estimations were shorter and displayed larger variations for normal- and large strides compared to the reference system. A closer inspection at the raw data revealed that the peaks of the acceleration data were cut off at ±4 g, which was the acceleration range for the sensors, as shown in Figure 8. It is highly likely that with the peak acceleration values missing, the double integration algorithm was not able to calculate the full length of the movement trajectory in affected dimensions, thus leading to a shorter estimation of the stride lengths.

It is interesting to see that for the MMR IMUs, the stride length estimations seemed to form two clusters in both the correlation plot and the Bland-Altman plot, as shown in Figure 9a,b. Results from the Move 4 IMUs are displayed in parallel as references. In order to identify the clusters, a K-Means clustering algorithm (number of clusters = 2, random state = 0) was applied to the stride length differences between IMU estimation and OptoGait measurement, and the mean silhouette coefficient ($S_{c}$) was calculated for both MMR and Move 4 IMU strides. $S_{c}$ = 1 means the sample is very far away from the neighboring clusters (i.e. clusters are very well separated), and $S_{c}$ = −1 means the sample is assigned to the wrong cluster. The mean $S_{c}$ for MMR strides was higher than that for the Move 4 strides, indicating that the two K Means clusters from MM strides are more distinct than those from Move 4 strides. The clustered strides are illustrated in Figure 9c,d. For MMR IMUs, the clusters identified by the K-Means method were consistent with the clusters of strides from left and right foot in Figure 9e. Whereas for Move 4 IMUs, the two clusters of strides did not resemble the clusters from the feet in Figure 9f. With the above analysis, it was confirmed that the two MMR IMUs from the left and right feet had inconsistent performances, and according to the data distributions in the regression plot as well as the Bland-Altman plot, it was apparent that MMR IMUs from both feet needed calibration.

The accuracy of stride length estimations were generally lower for all three types of strides recorded by the Portabiles IMUs compared with other IMUs. The correlation coefficient was 0.96, and RMSE was 0.11 m. The average limits of agreement of 0.21 m was larger than those from all other IMUs. Errors in IMU and OptoGait stride matching are unlikely to be the cause, as the stride time estimation showed a similar distribution and accuracy as all other IMUs. The IMUs may need more precise calibration, as the sample devices provided by the producers were only intended for initial testing, and the user was expected to perform a systematic calibration based on the procedures described by Ferraris et al. [40].

Similarly, data from the Shimmer IMUs also resulted in a larger variation of stride length accuracy for all three types of strides. The Shimmer IMUs were calibrated multiple times using the Shimmer 9-DoF calibration software. With the detailed user instructions and tips from the Shimmer customer support, the calibration parameters improved and stabilized over the repetitions. The correlation analyses and Bland-Altman plots for stride lengths from all IMUs can be found in the Appendix A (Figure A3 and Figure A4).

## 5. Discussion

### 5.1. General Comments on the Devices

In this section, general comments are provided for each device. The distinctive features, as well as advantages and disadvantages of these features to certain use cases, will be summarized.

Bonsai devices have a flexible visualization interface. In the iOS app, the user is allowed to choose signal modalities (e.g., raw acceleration data and quaternions) from arbitrary connected IMUs and visualize them in real time during data recording. The default acceleration range of ±4 g might be a limitation for certain use cases, but it is possible to customize the acceleration range to ±16 g depending on the signal modalities being recorded. Thus, the Bonsai devices are especially well suited when IMU signal visualization is needed.

MMR devices are the most cost-effective ones among all the brands being tested in this study. They provide multiple sensor modalities, and with the sampling rate of the accelerometer and gyroscope up to 1024 Hz, they are capable of capturing data in a wide range of use cases. The 8 MB memory significantly limits the onboard recording time, and instead of dedicated customer support, the user is often redirected to online tutorials or community forums (which nevertheless offers sufficient information) to solve their technical issues. Therefore, the MMR devices are a very good choice for researchers who work on method development topics with short recording sessions. The new generation of MMR (MMR+) devices also contain a vibration motor, which can provide haptic feedback to the user.

Portabiles devices are the only ones tested in this study that utilize wireless charging. This charging/connectivity option might be a significant advantage when working with elderly or certain disease populations, as in long-term studies, these subjects might experience difficulties handling Micro USB connections. Portabiles are relatively new on the market, researchers are advised to communicate with the device providers about use case specific requirements (data format, firmware update, SDKs, etc.).

Move 4 offers reliable sensor setup and data transfer via a desktop software. The recording protocol can also be configured, such as predefining the start and end of recording in real world time. Connecting and configuring the devices one by one with the software can be tedious with short recording sessions and rapid repetitions. However, with features such as the water proof level, onboard memory and battery capacity, as well as the smartphone questionnaire app (with which the researcher can predefine upon which sensor signal features the subject will get a prompt in the app, and answer corresponding questions), the Move 4 devices are specialized for long-term lifestyle tracking studies.

Gait Up devices offer reliable sensor setup and data transfer via Micro USB to the PC or Mac, and multiple data collection control options such as smartphone, smartwatch, or physical buttons on the device. The companion products from Gait Up support a set of routine gait, balance and motor function tests (e.g., 10 m walk, bipodal balance, or range of motion). The Micro USB connection can be challenging in home use cases for the elderly. However, according to the manufacturer, improvements are planned for the next generation of IMU (Physilog®6), which is scheduled to be released in 2020. The new IMUs will have USB C connection and have a larger battery capacity that promises more than double the maximum recording time of the current Physilog®5.

Less information is available in English about the EXLs3 devices. A Windows software is available for sensor configurations and data collection. Using the Bluetooth communication protocol offered by the company, the user can develop his or her own sensor configuration and data recording tools. Researchers interested in this product should get in touch with the device provider for more use case specific information.

Shimmer IMUs have a set of software and smartphone apps for sensor configuration and recording. The pro versions (with an annual subscription fee) offer a wide range of functionalities such as synchronization, data management, and algorithms. While the restrictions in basic versions make them more suited for single device recordings. The calibration software by Shimmer makes it easy to recalibrate and visualize the raw data after calibration, but it is crucial for the user to confirm the data quality after each calibration, as this could be a prominent error source for subsequent data analysis.

### 5.2. Selection of IMUs Depends on the Use Cases

To select an IMU from the large variety of commercially available products, it is important to first identify the research area and use case. A few typical setup options for consideration are described below.

For research topics that are related to algorithm development with a single or very few IMUs, it is common to have multiple short iterations of data collection in a controlled lab environment. In this case, battery life, data storage, and range of connection of most of the IMU devices tested in this study should be sufficient. Short-term recording is not likely to generate large amount of data, thus duration of data transfer via both Bluetooth or Micro USB should not be significantly different. Because of the exploratory nature of such studies, easy and fast configuration procedures for recordings (e.g., selecting sensor modalities and setting sampling rates) would be an advantage. One might eventually prefer to have more developer options in order to further customize the study workflow. For this purpose, IMUs such as MMR or Portabiles are suitable candidates. For example, an Android app has been developed using the API provided by MMR for the investigation of optimal IMU location on the foot for gait analysis [46].

An example use case with multiple IMUs for controlled lab environments could be full body motion tracking. In this case, apart from the considerations discussed above, one should also look into the configuration and synchronization procedures of the devices. It is no longer practical to have an application that connects to devices one by one, and the number of simultaneous device connections could be a limiting factor. The Xsens IMUs are very commonly used for human motion tracking. The Xsens MVN Analyze system is specialized for real-time full body motion tracking and 3D visualization [47]. The Xsens IMUs (MTw Awinda) were not tested in the current study because they operate almost exclusively in data streaming mode and have a limited range to their signal receiver. Therefore, the product did not meet our inclusion criteria for pervasive health use cases. The Xsens DOT IMUs released in early 2020 offers a light weight alternative to the MTw Awinda IMUs, and is worth testing for the lab-based use cases mentioned above.

Another intensively researched area of IMU application is using wearable devices for activity tracking in daily life. For this purpose, the battery life and data storage become an important factor for IMU selection. They should be sufficient for the entire recording session, which can last up to several weeks. The connectivity type during data recording are also crucial factors. Preferably, the device should be able to store data onboard without any external connections during recording, thus enabling the subject to move freely with their daily life activities. In such long-term studies, the subject normally will not wear too many sensors on the body, which reduces the constraints of multi-device configuration and synchronization. Additionally, there are usually lower requirements for sampling rate. Depending on anticipated data recording situations, waterproof devices may have be of advantage. The large amount of data collected during long-term recordings are transferred most efficiently via USB connections instead of Bluetooth. The Move 4 device is a good example that meets the above mentioned criteria, and has been used in studies where the subjects were monitored in the home environment [48].

Among the devices tested in the current study, Gait Up and Portabiles are the more versatile ones for various use cases. They generally meet all requirements for both controlled lab experiments and long-term real life recordings. It is worth emphasizing that the seven IMUs tested in the current study all have their advantages and disadvantages relative to the intended use case. The comparison results of the current study is a concrete example with a specific use case (i.e., gait analysis). Researchers should generalize the conclusions in our comparison with caution, since in a different use case, the comparison results are likely to be different.

### 5.3. Insights into Data Quality Provided by Use Case Algorithm

By comparing the gait parameters calculated from the same algorithm using data from different IMUs, the quality of IMU raw data could be magnified and visualized.

Regarding the comparison of stride times in our study, almost all tested IMUs showed similarly high accuracy. Whereas in comparison of the stride lengths, data from different IMUs lead to very different levels of accuracy compared to the gold standard gait analysis device. This suggested that with sufficient sampling rate and constant sampling intervals, temporal features that are extracted directly from raw IMU data are reliable. In the case of the current study, the gait events (initial contact and foot off) were extracted using a peak detection-based method directly from the raw angular velocity data. In contrast, the estimation of spatial features from IMU data is more challenging, as complex analysis procedures are often needed to extract these features. For example, the stride lengths were estimated using the principle of double integration and an error state Kalman filter. Similar trends were also found in other studies on IMU gait analysis [37]. Issues with calibration could also cause inaccurate analysis results, as data from IMUs that are known to require custom calibration (MMR, Portabiles, and Shimmer) generally produced less accurate stride length estimations. Besides the general accuracy of the stride length estimations, larger errors can be observed for large strides in the Bland–Altman Plots compared to normal and small strides. This finding was especially true for the Bonsai IMUs, whose data was partially cut off by the acceleration range at ±4 g. However, similar issue was also observed for several of the other IMUs (Portabiles, Move 4, Gait Up, and EXLs3), where the estimations for small and normal strides displayed higher levels of agreement with the Optogait system. Therefore, it is recommended that data with varying features (e.g., in our case stride length) should be used for validation, in order to capture the behavior of the tested IMUs in a more comprehensive way. When using the IMUs for data collection, researchers should also be aware that the reliability of the analysis results might differ in special cases.

It is worth noting that results from the raw data exploration (sampling intervals, baseline accelerometer and gyroscope values) did not strongly correlate with the performance of the IMUs in the gait analysis test. This has two implications: (1) to understand the error characteristics and the calibration status of the IMUs, more sophisticated calibration procedures are needed. Such analysis is beyond the scope and purpose of the current study, but a commonly used IMU calibration method is described by Ferraries et al. [40] (2) the IMUs should be tested on the intended use cases. This is a straightforward and necessary strategy to find out which IMU should be selected for a certain study. For example, the Shimmer IMUs recorded among the most accurate values in our initial tests with the sampling interval and baselines, but their data did not produce the best results for gait analysis. However, this does not prevent the Shimmer IMUs from outperforming other IMUs when used for different algorithms or use cases.

The gait analysis algorithm in the current study should merely be regarded as one example of use case testing. It is important for researchers to also determine the underlying factors that are affecting the outcome of their studies. IMU selection is a starting point, mechanism of fixing the IMUs, and body location for data collection are examples of the next considerations [49].

## 6. Conclusions

The current study proposed a guideline for the selection of IMUs for pervasive healthcare-related use cases and compared the specification, data collection procedures, as well as the data quality of seven consumer grade IMUs. We showed that it is necessary to consider the features of the IMUs with respect to areas of research. Figure 2 and Table 1, Table 2 and Table 3 in this paper summarize these points. Exploratory analysis of the raw data revealed differences between the IMUs in terms of the type and quality of data they provide. By comparing spatio-temporal gait parameter estimates from the IMU data, we demonstrated that testing the IMUs for their intended use may be necessary and highlighted the importance of proper sensor calibration to data analysis results.

As an example IMU selection, we presented device information and data processing results for a selection of popular commercial IMUs, and evaluated the performance of the tested IMUs with a typical gait analysis algorithm. For researchers who are working on similar topics and interested in IMUs tested in the current study, our findings are directly relevant. For a broader audience, our findings are generally informative, and the guidelines proposed for IMU selection can be generalized to new IMU models and new algorithms. As the IMU market is growing fast, researchers are encouraged to go beyond the scope of this paper in search of the most suitable IMU for their use cases.

## Figures and Tables

**Figure 1 sensors-20-04090-f001:**
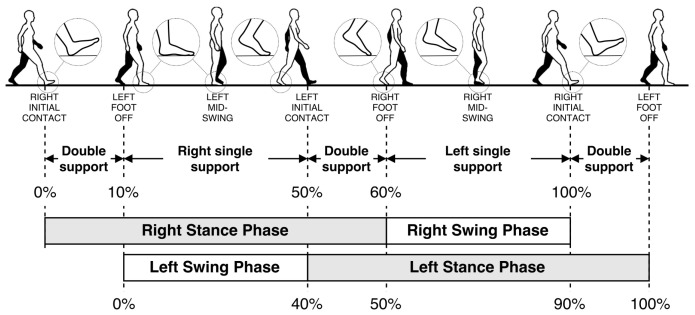
Human gait cycle. Reprinted from “Inertial Sensor-Based Robust Gait Analysis in Non-Hospital Settings for Neurological Disorders” by Tunca, C.; Pehlivan, N.; Ak, N.; Arnrich, B.; Salur, G.; Ersoy, C., 2017, Sensors (Switzerland), 17, 1–29. © 2017 by the authors. Reprinted with permission.

**Figure 2 sensors-20-04090-f002:**
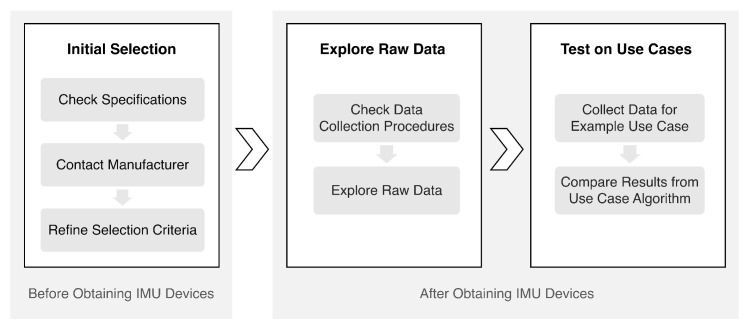
Overview of the proposed guideline for inertial measurement unit (IMU) selection.

**Figure 3 sensors-20-04090-f003:**
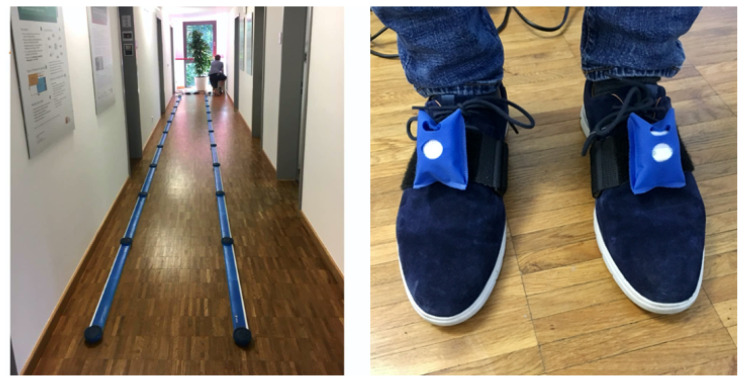
Gait data recording set-up. Left: the 10 m OptoGait walkway. Right: the IMUs were fixed on top of the shoes.

**Figure 4 sensors-20-04090-f004:**
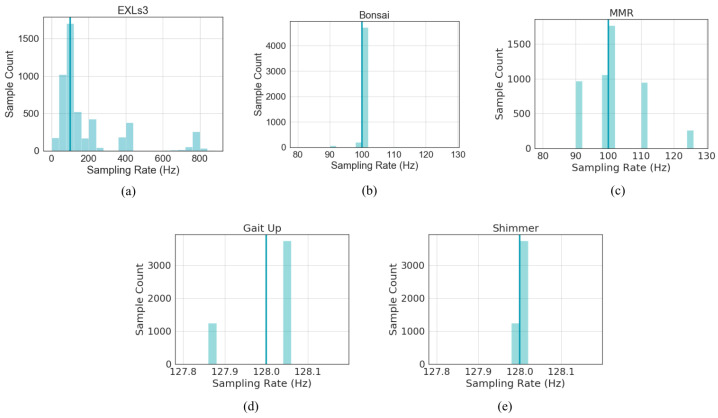
Sampling rates for (**a**) EXLs3 IMU, (**b**) Bonsai IMU, (**c**) MMR IMU, (**d**) Gait Up IMU, (**e**) Shimmer IMU. Sampling rates were calculated from timestamp intervals for the IMUs that provide timestamps. Configured sampling rates for the recording marked by a turquoise line.

**Figure 5 sensors-20-04090-f005:**
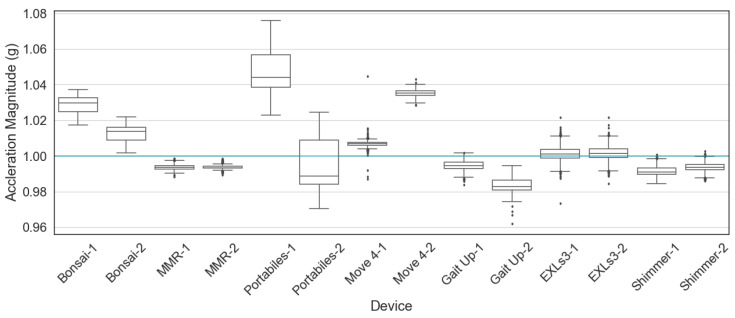
Accelerometer baselines. Acceleration at 1 g (gravity) marked by a turquoise line.

**Figure 6 sensors-20-04090-f006:**
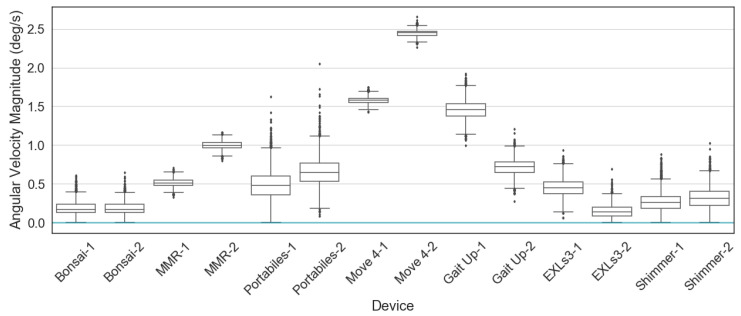
Gyroscope baselines. Angular velocity at 0 deg/s marked by a turquoise line.

**Figure 7 sensors-20-04090-f007:**
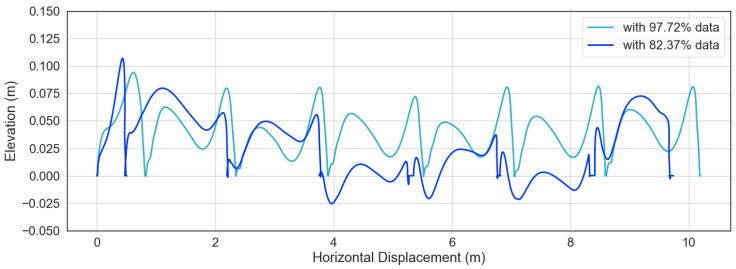
Disruption of the estimated stride trajectories due to data loss during Bluetooth transmission from the EXLs3 IMUs. Each line is a lateral view of the displacement of one foot for a walking subject.

**Figure 8 sensors-20-04090-f008:**
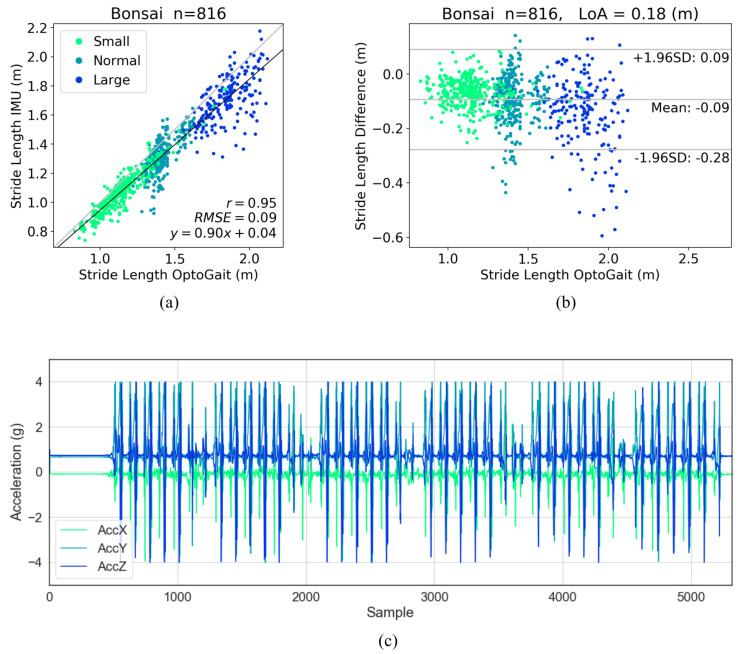
Stride length evaluation for the Bonsai IMUs and example raw acceleration data. (**a**) correlation analysis of stride length, (**b**) Bland-Altman plot of stride length, stride lengths of longer strides were underestimated, and (**c**) raw acceleration data from one example recording session, signals were cut off at the ±4 g limit.

**Figure 9 sensors-20-04090-f009:**
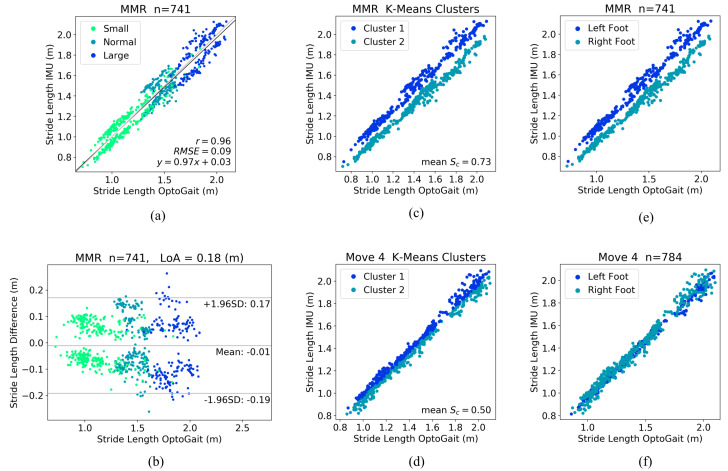
Evaluation of stride lengths from left and right foot derived from MMR IMU data, and reference data from Move 4 IMUs. (**a**) and (**b**) correlation analysis and Bland-Altman plot of stride length derived from MMR data, (**c**) and (**d**) K-Means clustering based on the stride length differences between IMU and OptoGait measurements, (**e**) and (**f**) correlation analysis of stride length grouped by left and right foot, for MMR and Move 4 IMUs, respectively. $S_{c}$: silhouette coefficient.

**Table 1 sensors-20-04090-t001:** Recommended specifications for filtering the IMUs of interest.

Specifications	Relevance
Dimensions	Unobtrusiveness
Accelerometer Range	Movements of interest (e.g., high/low speed)
Gyroscope Range	
Max. Sampling Rate	
Onboard Memory	Long-term recording/daily life monitoring
Battery Capacity	
Additional Sensors	
Charging Options	
Water/Dust Proof	
Onboard Sensor Fusion	Quick demo
Synchronization (multi-sensor control)	Convenience for data recording & processing
Sensor Configuration and Data Collection Control	
Calibration Options	High precision analysis
Developer Options	Use case customization
Special Features	

**Table 2 sensors-20-04090-t002:** Recommended aspects of data collection for filtering IMUs.

Aspects	Relevance
Start & End Recording Control	Flexibility and reliability of recordings
Data Readout	
Download required after each recording	Workflow and time consumption
Timestamps	High precision data analysis
Max. Recording Limitation	Long-term recording/smart home monitoring
Max. Recording Time	

**Table 3 sensors-20-04090-t003:** Selection criteria and example of excluded IMU devices.

Criteria	Example of Excluded Devices
Is an integrated commercial IMU (with IMU sensors, battery, memory, and Bluetooth communication modules)	BMI160 (Bosch, Gerlingen, Germany) and MPU6050 (InvenSense, San José, CA, USA)
Has at least accelerometer and gyroscope	ActiGraph wGT3X-BT (ActiGraph, Pensacola, FL, USA)
Able to record data onboard	MTw Awinda (Xsens, Enschede, The Netherlands)
Allows raw IMU data access	Steel HR (Withings, Issy-les-Moulineaux, France) and MiBand (Xiaomi, Hangzhou, China)
Appears affordable (below 600 €)	OPAL (APDM, Portland, OR, USA), Blue Trident IMU (Vicon, Oxford, UK), and Perception Neuron Studio Inertial System (Noitom, Miami, FL, USA)

**Table 4 sensors-20-04090-t004:** IMU devices tested in this study.

Device Name Used in This Paper	Official Device Name	Company	Sampling Rate (Hz)
Bonsai IMU	QuantiMotion	Bonsai Systems, Zurich, Switzerland	100
MMR IMU	MetaMotionR	MbientLab, San Francisco, CA, USA	100
Portabiles IMU	NilsPod	Portabiles, Erlangen, Germany	102.4
Move 4 IMU	Move 4	movisens, Karlsruhe, Germany	128 *
Gait Up IMU	Physilog®5	Gait Up, Lausanne, Switzerland	128
EXLs3 IMU	EXL-s3	EXEL, Bologna, Italy	100
Shimmer IMU	Shimmer 3	Shimmer Research, Dublin, Ireland	128

* Downsampled from 256 Hz.

**Table 5 sensors-20-04090-t005:** Subject characteristics for each of the tested IMU.

	Sex	Age (Years)	Body Height (cm)	BMI (kg/m${}^{2}$)
Bonsai	3 Females, 2 Males	25.6±2.9	171.6±8.6	21.8±2.7
EXLs3	2 Females, 2 Males	24.5±1.7	174.5±6.6	21.6±3.0
Gait Up	1 Female, 4 Males	25.2±2.2	182.2±10.1	23.0±2.5
MMR	3 Females, 2 Males	24.4±2.5	172.8±4.4	21.0±2.8
Move 4	2 Females, 3 Males	25.2±2.2	177.4±8.6	22.6±3.3
Portabiles	1 Female, 4 Males	25.8±2.8	180.0±7.8	22.9±2.5
Shimmer	4 Females, 1 Male	27.2±2.4	172.0±10.3	21.6±3.6

BMI: Body mass index, BMI = Body weight (kg)/[Body height (m)]${}^{2}$.

**Table 6 sensors-20-04090-t006:** IMU device specifications.

Specifications		Bonsai	MMR	Portabiles	Move 4	Gait Up	EXLs3	Shimmer
Dimensions (mm)		36.5 × 32.0 × 13.5	36 × 27 × 10	28 × 23 × 11.5	62.3 × 23 × 11.5	47.5 × 26.5 × 10	54 × 33 × 14	51 × 34 × 14
Onboard Memory		32 MB	8 MB	250 MB	4 GB	8 GB	1 GB	8 GB
Battery Capacity		250 mAh	70–100 mAh	120 mAh	380 mAh	140 mAh	200 mAh	450 mAh
Max. Sampling Rate ^1)^		100 Hz	800 Hz	1024 Hz	64 Hz ^4)^	512 Hz	200 Hz	1024 Hz
Accelerometer Range	±16 g	● ^2)^	●	●	●	●	●	●
Gyroscope Range	±2000 deg/s	●	●	●	●	●	●	●
Additional Sensors	Magnetometer	●	●				●	●
	Barometer		●	●	●	●		
	Altimeter							●
	Temperature Sensor		●	●	●	●		
	Ambient Light Sensor		●					
Charging Options	Micro USB	●	●		●	●	●	●
	Additional Adaptor / Dock				●		●	●
	Wireless			●				
Waterproof (IP64)					●	●		
Calibration Options	Calibration Software							●
	Calibration Status	●	● ^3)^					
Developer Options	Example Scripts for Bluetooth Communication	●			●			
	Javascript API		●					
	Java API		●		●			●
	Swift API		●					
	Python API		●			●		
	Matlab Instrument Driver					●		●
	LabVIEW Instrument Driver							●
	C# / C++ API		●			●		●
Other Features	Raw Data Visualization	●		●		●	●	●
	Onboard Sensor Fusion	●	●			●	●	● ^5)^
	Orientation Visualization	●				●	●	● ^5)^
	Other	Flexible live visualization of the signal modalities	Online user forum and tutorials		Smartphone questionnaire app, vibration alarm	Gait, balance and motor function tests		

Information in this table is accurate as of 21st July 2020. ^1)^ With accelerometer and gyroscope. ^2)^ Acclerometer range can be customized to ±16 g. ^3)^ Calibration status only displayed in custom apps. ^4)^ Max. sampling rate can be customized to 256 Hz. ^5)^ Only available for paid software / app with annual subscriptions.

**Table 7 sensors-20-04090-t007:** Data collections procedures.

Specifications		Bonsai	MMR	Portabiles	Move 4	GaitUp	EXLs3	Shimmer
Sensor Configuration	Android (Smartphone)		●	●		●		●
	Android (Smartwatch)					●		
	iOS	●	●					
	Windows				●	●	●	●
	Mac					●		
Start & End Recording	Android (Smartphone)		●	●		●		●
	Android (Smartwatch)					●		
	iOS	●	●					
	Windows				●		●	●
	Physical Buttons on Device					●		●
	Multi Device Control	●	●	●		●	●	● ^1)^
Download required after recording			●		●			
Data Readout	Bluetooth Download in App	●	●	●		● ^2)^		
	USB Download to PC				●	●	●	●
	USB Download to Mac					●		
Timestamps	Timestamps Recorded	●	●			●	●	●
	Unix Time		●				●	●
	Start from Zero	●				●		
	Synchronized Timestamps	● ^3)^				●	● ^3)^	● ^1)^
Max. Recording Limitation	Battery Capacity				●	●	●	●
	Memory Size	●	●	●				
Max. Recording Time (h)	Device 1	2.23	0.72	45.50	~168 ^4)^	13.38	~3 ^4)^	36.49
	Device 2	2.23	0.73	45.50	~168 ^4)^	12.65	~3 ^4)^	40.76

Information in this table is accurate as of 21st July 2020. ^1)^ Only available for paid software / app with annual subscriptions. ^2)^ BIN files can be downloaded to the internal memory of the phone. ^3)^ Synchronized timestamps: not exactly the same values, but start- and end of recordings roughly synchronized. ^4)^ Max. recording times were not experimentally confirmed.

**Table 8 sensors-20-04090-t008:** Evaluation of stride lengths derived from IMU data with correlation analysis and Bland-Altman Plot.

Analysis	Parameters	Bonsai	MMR	Portabiles	Move4	Gait Up	EXLs3	Shimmer
Sample Size	Num. of Strides	816	741	778	784	783	612	821
Correlation Analysis	*r*	0.95	0.96	0.96	0.99	0.99	0.99	0.97
	Slope	0.90	0.97	0.97	1.01	1.00	0.96	0.91
	Intercept (m)	0.04	0.03	0.02	−0.05	−0.04	0.04	0.11
	RMSE (m)	0.09	0.09	0.11	0.04	0.04	0.04	0.08
Bland-Altman Plot	LoA (m)	0.18	0.18	0.21	0.08	0.09	0.09	0.16

r: correlation coefficient, RMSE: root mean square error, LoA: average limits of agreement at ±1.96 standard deviation.

**Table 9 sensors-20-04090-t009:** Evaluation of stride times derived from IMU data with correlation analysis and Bland-Altman Plot.

Analysis	Device	Bonsai	MMR	Portabiles	Move4	Gait Up	EXLs3	Shimmer
Sample Size	Num. of Strides	816	741	778	784	783	612	821
Correlation Analysis	*r*	0.97	0.95	0.95	0.95	0.94	0.94	0.97
	Slope	0.99	0.98	0.99	1.00	0.99	0.97	0.99
	Intercept (s)	0.02	0.03	0.01	0.00	0.02	0.04	0.01
	RMSE (s)	0.05	0.06	0.05	0.04	0.04	0.04	0.06
Bland-Altman Plot	LoA (s)	0.09	0.11	0.09	0.08	0.08	0.09	0.11

r: correlation coefficient, RMSE: root mean square error, LoA: average limits of agreement at ±1.96 standard deviation.

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
