# Peer review of "How We Found Our IMU: Guidelines to IMU Selection and a Comparison of Seven IMUs for Pervasive Healthcare Applications"

_sensors, 2020, doi:10.3390/s20154090_

Round 1

Reviewer 1 Report

Brief summary: The aim of the present work is twofold: (1) to propose guidelines for the correct selection of IMUs for healthcare applications; (2) to apply these guidelines for the comparison of seven types of IMUs in terms of specifications, data collection procedures and data quality. First, general guidelines for IMUs selection are defined with three major steps. Then, gait data of five healthy young subjects are collected simultaneously with IMUs and a gold standard. Different walking sessions are performed in order to test seven types of IMUs. Overall, results demonstrate that the proposed guidelines can be generalized also to applications different from gait. In addition, since the tested IMUs reveal differences in terms of type and quality of provided data, their features have to be considered with respect to purpose of research.         

Broad comments: The topic of the present work is certainly interesting due to the constantly growing attention paid to Inertial Measurement Units. The title is relevant and completely informative of the content. Moreover, the introduction is well-structured and detailed. Indeed, literature knowledge about healthcare applications, and especially about gait analysis, is clear. In addition, the research question is clearly outlined, by stressing the literature gap about the difficulty in selecting the suitable IMUs for a specific application. Consequently, according to what is already known about the topic, the research question seems to be justified. Furthermore, both the proposed guidelines for IMUs selection and the description of included and excluded devices are detailed and clear. Finally, tables and figures are relevant and clearly presented.

However, many parts of the paper need to be improved for a better comprehension:

  • The background context, the gap in knowledge addressed by the paper and the aim of the work are clear. However, methods and results are not clearly presented. Indeed, no reference is made to subjects’ selection, data collection during gait and data elaboration. Finally, key conclusions could be improved to highlight the new knowledge introduced by the paper.
  • Materials and Methods. In the description of preprocessing, authors talk about raw data files without having previously introduced how they obtained these files. Subject selection is not described, because age, anthropometric data and other characteristics of participants are missing. Furthermore, authors do not refer to any informed consent obtained from subjects, which represents a key point for the correct conduct of a research. In the description of the experimental protocol, it is not clear if the three testing sessions were repeated by all subjects for each of the seven tested types of IMUs. In addition, details about the synchronization between IMUs and the gold standard, the software used for conducting data analysis, the detection of gait events and the assessment of spatio-temporal parameters are missing. The study could be considered valid and reliable, but general improvements in the description of materials and methods are necessary to make it reproducible.
  • Overall, the section of results is enriched with comments typical of a discussion section. Since this aspect can be confusing to the reader, this part needs to be restructured by inserting only figures and tables related to obtained results. In particular, the subsection with general comments on devices should be moved in the discussions because it represents an interpretation of results.
  • Discussions and conclusions. Even if discussions are well developed, they could be improved by inserting some references to support findings. Furthermore, conclusions need a clearer generalization of results about the performed comparison of the seven types of IMUs.
  • References are relevant, recent and correctly referenced. However, some major references related to the use of wearable devices for healthcare applications should be included:
    • Zijlstra, W. (2004). Assessment of spatio-temporal parameters during unconstrained walking. European journal of applied physiology, 92(1-2), 39-44.
    • Ghislieri, M., Gastaldi, L., Pastorelli, S., Tadano, S., & Agostini, V. (2019). Wearable Inertial Sensors to Assess Standing Balance: A Systematic Review. Sensors, 19(19), 4075.
    • Bertoli, M., Cereatti, A., Trojaniello, D., Avanzino, L., Pelosin, E., Del Din, S., ... & Hausdorff, J. M. (2018). Estimation of spatio-temporal parameters of gait from magneto-inertial measurement units: Multicenter validation among Parkinson, mildly cognitively impaired and healthy older adults. Biomedical engineering online, 17(1), 58.
  • Many sentences should be shortened to improve the clarity of the paper content.

Specific comments:

  1. Line 4. End the sentence with the word “procedures” and start a new sentence with “However,…”.
  2. Line 18. Specify that the presence of a magnetometer defines a MIMU instead of a simple IMU.
  3. Line 23. Specify that the healthcare application proposed in this paper is related to gait.
  4. Line 24. Modify the sentence with “…can contribute to the knowledge of IMUs selection for…”.
  5. Line 92. Did you mean “stance/swing time” instead of “swing/time”?
  6. Line 93. Figure 1 does not illustrate the relationship among spatio-temporal parameters, but it shows the gait cycle, its major events and its phases.
  7. Lines 102-105. Improve this sentence, because it is too long and it has too many verbs.
  8. Line 135. The term “fact-checking” is not so clear.
  9. Lines 145-146. Support this sentence with a reference.
  10. Lines 154-157. Divide this sentence because it is too long.
  11. Lines 157-160. Divide this sentence because it is too long.
  12. Lines 171-173. If it exists, consider inserting a reference with this synchronization procedure as an example.
  13. Line 188. Modify “…that example devices be obtained…” with “…that example devices are obtained…”.
  14. Lines 198-200. Divide this sentence because it is too long.
  15. Lines 200-202. Improve this sentence, because it is too long and it has too many verbs.
  16. Line 216. A new paragraph is not necessary.
  17. Lines 232-234. Divide this sentence because it is too long.
  18. Lines 259-261. The threshold inserted in Table 3 for the distinction between high and low cost should be also inserted in the text.
  19. Lines 304-305. Also include the definition of ti-1.
  20. Lines 322-325. This part would be more appropriate in the section of data analysis.
  21. Line 331. Do not start the sentence with “And…”.
  22. Line 353. Figure 3 was already introduced at line 334.
  23. Table 3. Improve the first column by inserting the subject “IMU” for each row.
  24. Line 365. Specify with respect to which reference system the matrix Rt is evaluated.
  25. Line 369. Specify that velocity and displacement are estimated at time t starting from their values at time t-1.
  26. Lines 381-382. Also define the meaning of Ft.
  27. Lines 387-388. Also define the meaning of Kt.
  28. Line 500. Substitute “were calculated” with “was calculated”.
  29. Line 507. Do not start the sentence with “And…”.
  30. Table 7 and Table 8. Measurement units are missing.
  31. Figures 8 and 9. Remove comments about results in the captions of these two figures.
  32. Line 568. The sentence ends with “:”, but it is not followed by a list of elements.

Reviewer 2 Report

The manuscript presents guidelines to select IMUs for persuasive healthcare applications. In addition, the authors have compared 7 well-known IMUs. The paper is well-written and describes clearly the main features of the existing IMUs and the proposed guidelines. However, in my opinion, some points of the comparison need to be clarified:

- I think that the correlation analysis (see Section 4.2) requires a detailed description of the parameter used (see Tables 7 and 8) and a deeper analysis about the results (i.e., Fig. 8, Fi. 9 and Appendix A include a lot of results that have not been analysed). I suggest the authors to extent Sections 4.2 and 5 in order to provide comprehensible (and practical) information obtained from the results.

- The authors determine the existence of clusters only with visual inspection. I think that it would be a good idea to use some clustering validation indices.

There are some typos that should be corrected:

- "NETLAB, Department of Computer Engineering" and "Division of Training and Movement Sciences" have the tag 2. One of them should have the tag 3.

- "the results of the use case algorithm testing depends [instead of depend] both"

Reviewer 3 Report

This article is written very clearly and comprehensibly. It is very useful for practice. I recommend it for publication without further modification.

Author Response

Dear Reviewer #3,

we thank you for your positive response and affirmative remarks. Since modifications were suggested, we attached the latest version of the manuscript for your revision.  

Best regards,

Lin Zhou on behalf of all co-authors

Round 2

Reviewer 1 Report

Reviewer believes the manuscript has been significantly improved and now warrants publication in Sensors.

This manuscript is a resubmission of an earlier submission. The following is a list of the peer review reports and author responses from that submission.

Round 1

Reviewer 1 Report

- The current article presents a guidelines for selecting and choosing an IMU for a given application in domain of Pervasive Healthcare Applications. This work apply this guideline to compare 7 commercial IMU, by considering their efficiency in spatio-temporel gait parameters of human walking patterns.   

It is an useful and a necessary kind of work, I particularly appreciate the practical and useful aspect of this work. 

- The article is very clear, well written and easy to read. The organization is very good and I particularly appreciate the quality of figures and Tables. 

  However, I have an overall comment about usability of this guideline for future works and researchers. Indeed this guideline suggest that the choice depends on the application (application-oriented choice) and methods used to get results used for comparison. On the other hand, this method suggest to be helpful for choosing the « right » IMU before having it. So how to chose the right IMU for the right application before?

There is only few points which requires an improvement/clarification as summarized below: 

- You separate the IMU and its efficiency from the methods used for getting the spatio-temporal gait parameters, since there is one chosen method used here and somehow it has an influence win the comparison. Is it possible that others methods gave different results in terms of comparison, by making some IMU close or better than currently considered as best IMU? I mean, this results of spatio-temporal gait parameters estimation depends on combination of IMU and the method.

- I am surprised that there is any IMU product from Xsens company. I am not sur about the price, but it is usually know as a reference product for this kind of application. 

- the sampling rate are around 100-128 Hz, isn’t there any product with more frequency, specially for application that exploit the acceleration data (for foot clearance estimation based on double integration) which requires a higher frequency? I know there is one example of product (STT systems) that have 400Hz and which is affordable. 

- It is not specified if the more sophisticated calibration of IMU is done here or not? also, the software included with IMU may change the raw data with some routines, since (for display), it the fact to use the raw data, without modification, ensure a good quality thanks to a calibration ?

 - Keyword 1 have to be removed in Keywords section.  

Reviewer 2 Report

This work presents a comparison between seven different commercial measurement units. The comparison focuses on the temporal parameters of gait and it is not clear if other aspects, such as the spatial parameters or the joint kinematics, are also comparable between different IMU systems and the optical measurement system.

A major concern is about the applicability of the presented work for other groups. A problem in the generalization of the results is the restriction in the devices selection. For instance, there are other sensors, either sold for biomechanical applications (ShadowMotion, Perception Neuron) or as independent IMUs (as the Bosch units such as BMI055, or VectorNav).

The scope of the paper should clarify which sensors have been considered for the analysis and detail the inclusion and exclusion criteria.

The Authors seem to have neglected some  relevant bibliography, for instance:

Al-Amri, Mohammad, et al. "Inertial measurement units for clinical movement analysis: Reliability and concurrent validity." Sensors 18.3 (2018): 719.

Reviewer 3 Report

The current study proposed a guideline for the selection of IMUs for pervasive healthcare related use cases and compared the specification, data collection procedures, as well as the data quality of seven consumer grade IMUs.

1.In section 3, theoretical or mathematical methods are considered to provide, not only a procedure.  

2. The formulas of calculation methods for test data should be given accordingly.

Reviewer 4 Report

The article does not present any new methodology or method. The article presents the results of one of the selected algorithms (why this one?) for estimating gait parameters based on signals measured by various sensors. Currently achieved results based on IMU signals are based primarily on properly selected and parameterized algorithms, which consists of many elements (also low-level): bias estimation, compensation of magnetometer disturbances, compensation of temperature influence, etc. Even based on sensors with poor parameters, good results, at a high level of abstraction (just like gait parameters), can be obtained. The novelty and meaning of publication are insignificant. Soon, other sensors and better algorithms will be available on the market. In my opinion, more significant can be a general methodology for comparison sensors (for example, on low-level using Allan's variance).

Round 2

Reviewer 3 Report

There is no other questions.

Reviewer 4 Report

I appreciate the work and changes made to the article. My concerns about the lack of general methodology and a very narrow and specific application of the results have been confirmed by numerous descriptions of research limitations. The results concern only to the tested one known algorithm on 7 selected sensors without examining how actually mentioned known problems of IMU signal measurement affect the results. How can this be extended to general methodology and wider application?
Other comments:
Excluding nowadays most-popular sensor in biomechanics is problematic (Xsens).
Why recording onboard data is so important?

My remark regarding the research methodology.
Does the lack of the same calibration procedure affect the results? If low-level signals are analyzed, why was not one calibration procedure carried out?
Line 181 – “It is common practice to ask colleagues for user experience and decide on the IMU accordingly” – On what basis is such a statement. In my opinion, this is an erroneous statement, there is already a lot of publications describing this problem.